# Structural Evaluation of a Nitroreductase Engineered for Improved Activation of the 5-Nitroimidazole PET Probe SN33623

**DOI:** 10.3390/ijms25126593

**Published:** 2024-06-15

**Authors:** Abigail V. Sharrock, Jeff S. Mumm, Elsie M. Williams, Narimantas Čėnas, Jeff B. Smaill, Adam V. Patterson, David F. Ackerley, Gintautas Bagdžiūnas, Vickery L. Arcus

**Affiliations:** 1School of Biological Sciences, Victoria University of Wellington, Wellington 6012, New Zealand; abby.sharrock@vuw.ac.nz (A.V.S.);; 2Wilmer Eye Institute, Johns Hopkins University, Baltimore, MD 21287, USA; jmumm3@jhmi.edu; 3Institute of Biochemistry, Life Sciences Center at Vilnius University, Saulėtekio Av. 7, LT-10257 Vilnius, Lithuania; narimantas.cenas@bchi.vu.lt; 4Auckland Cancer Society Research Centre, School of Medical Sciences, The University of Auckland, Auckland 1142, New Zealand; j.smaill@auckland.ac.nz (J.B.S.); a.patterson@auckland.ac.nz (A.V.P.); 5Te Aka Mātuatua School of Science, University of Waikato, Hamilton 3240, New Zealand; vic.arcus@waikato.ac.nz

**Keywords:** type I nitroreductase, NfsB, crystal structure, prodrug, theranostic imaging, SN33623, CB1954, metronidazole, cancer gene therapy, targeted cellular ablation

## Abstract

Bacterial nitroreductase enzymes capable of activating imaging probes and prodrugs are valuable tools for gene-directed enzyme prodrug therapies and targeted cell ablation models. We recently engineered a nitroreductase (*E. coli* NfsB F70A/F108Y) for the substantially enhanced reduction of the 5-nitroimidazole PET-capable probe, SN33623, which permits the theranostic imaging of vectors labeled with oxygen-insensitive bacterial nitroreductases. This mutant enzyme also shows improved activation of the DNA-alkylation prodrugs CB1954 and metronidazole. To elucidate the mechanism behind these enhancements, we resolved the crystal structure of the mutant enzyme to 1.98 Å and compared it to the wild-type enzyme. Structural analysis revealed an expanded substrate access channel and new hydrogen bonding interactions. Additionally, computational modeling of SN33623, CB1954, and metronidazole binding in the active sites of both the mutant and wild-type enzymes revealed key differences in substrate orientations and interactions, with improvements in activity being mirrored by reduced distances between the N5-H of isoalloxazine and the substrate nitro group oxygen in the mutant models. These findings deepen our understanding of nitroreductase substrate specificity and catalytic mechanisms and have potential implications for developing more effective theranostic imaging strategies in cancer treatment.

## 1. Introduction

Gene-directed enzyme-prodrug therapy (GDEPT) is a precision medicine strategy based on the vector-mediated delivery of a genetically encoded prodrug-converting enzyme to tumor cells [1,2,3]. Systemic administration of the prodrug then results in preferential activation of the corresponding toxin in tumors. Several enzyme-prodrug combinations have been tested in human trials, most notably herpes simplex virus thymidine kinase with ganciclovir [4,5], cytosine deaminase with 5-fluorocytosine [6,7], and the *Escherichia coli* nitroreductase NfsB with 5-(aziridin-1-yl)-2,4-dinitrobenzamide (CB1954) [8,9] (Figure 1). The former two systems employ nucleoside analog prodrugs that, upon activation, are incorporated into replicating DNA, whereas the nitroreduction of CB1954 yields species that form DNA mono-adducts or cross-links independent of replication [10,11]. As such, NfsB/CB1954 offers the potential advantage of targeting quiescent as well as actively replicating tumor cells. Phase I/II clinical trials showed promise, but concluded that native NfsB was insufficiently active at achievable plasma concentrations of CB1954, making heightened activity desirable [9]. It has also been widely recognized that non-invasive imaging modalities, such as positron emission tomography (PET) that can report on vector localization in vivo, are needed to support the clinical progression of nitroreductase GDEPT strategies [12,13,14,15].

We and others have shown that 2-nitroimidazole PET-capable probes, originally designed to image tumor hypoxia through conversion to a cell-entrapped form by endogenous oxygen-sensitive nitroreductases, can potentially be repurposed for the theranostic imaging of vectors labeled with oxygen-insensitive bacterial nitroreductases [12,13,16]. However, we observed that *E. coli* NfsA is substantially more active with 2-nitroimidazole substrates than NfsB [13,16]. To enable improved imaging of NfsB, two of us (J.B.S. and A.V.P.) designed SN33623, a 5-nitroimidazole analog of hypoxia probe EF5 [2-(2-nitro-1*H*-imidazol-1-yl)-*N*-(2,2,3,3,3-pentafluoropropyl)-acetamide] [17] (Figure 1). Not only is this 5-nitroimidazole analog a superior substrate for NfsB, its lowered one-electron transfer potential relative to EF5 also made it less responsive to tumor hypoxia. We subsequently performed sequence-activity analyses for 12 phylogenetically related NfsB enzymes and identified two residue substitutions (F70A/F108Y) that, when introduced into NfsB, improved activity with SN33623 by at least ten-fold relative to unmodified NfsB [17]. Serendipitously, these substitutions also substantially improved activity with CB1954 and metronidazole (Figure 1), the latter being of interest for targeted cell ablation applications in transgenic zebrafish models of disease [18] as well as a possible biosafety agent for the elimination of nitroreductase-encoding vectors post-GDEPT [13].

As the mechanistic basis for these activity improvements was uncertain, we sought to obtain a crystal structure of *E. coli* NfsB F70A/F108Y and identify the contributions of these residue substitutions to improved activity with each nitroaromatic substrate. Here, we describe our solution of the *E. coli* NfsB F70A/F108Y structure to 1.98 angstroms and comparative modeling of each of SN33623, CB1954, and metronidazole in the active sites of the parental and F70A/F108Y enzymes.

## 2. Results and Discussion

### 2.1. Crystal Structure of the Double-Mutant F70A/F108Y

*E. coli* NfsB belongs to an extensive, functionally diverse nitroreductase superfamily, members of which are united by a conserved structural fold and fundamental mechanism of catalysis [19]. NfsB family nitroreductases catalyze the reduction of nitroaromatic compounds, a step that is pivotal in various biochemical processes and exploited for bioremediation and cancer therapies [20]. This catalytic mechanism involves hydride transfer from a reduced cofactor, such as reduced nicotinamide adenine dinucleotide (phosphate) (NAD(P)H), to the nitro group of the substrate, leading to its conversion via a nitroso intermediate into the corresponding hydroxylamine or amine. The mechanism follows a ping-pong bi-bi kinetic model, involving two redox half reactions. The reductive half reaction consists of initial hydride transfer from NAD(P)H to a bound flavin mononucleotide (FMN) moiety, and the oxidative half reaction consists of subsequent hydride transfer from the reduced FMN to an electron acceptor substrate (e.g., the nitro group of an aromatic ring; Appendix A) [21]. Several crystal structures of *E. coli* NfsB and its mutants have previously been reported [22,23,24,25,26]. Informed by these studies, the crystal structure of *E. coli* NfsB F70A/F108Y was resolved.

*E. coli* NfsB F70A/F108Y was successfully expressed, purified, and crystallized using a sodium chloride, sodium acetate, and polyethylene glycol precipitant, resulting in crystals that diffracted to 1.87 Å (Figure 2A). The mutant crystal belongs to the orthorhombic space group P 2_1_ 2_1_ 2_1_ with unit cell parameters a = 87.22, b = 95.59, c = 112.31, α = 90°, β = 90°, and γ = 90°. The asymmetric unit contained four molecules, with a Matthews coefficient of 2.45 and a solvent content of 49.97%. Coordinate files and structure factors for the full structure have been deposited in the Protein Data Bank with PDB code 8V5B. The data collection and refinement statistics are summarized in Table 1.

The biological assembly of *E. coli* NfsB F70A/F108Y is a homodimer of two 24 kDa subunits, where each monomer exhibits the conserved α+β fold and FMN binding residues characteristic of members of the nitroreductase superfamily [19]. On comparison of the four monomers in the asymmetric unit, there are minor differences between chains (the root mean square deviation (RMSD) of Cα atoms is <0.15 Å), potentially resulting from crystal development. The structure of the double mutant in its dimeric form is shown in Figure 2B. Each monomer has a total of 13 alpha helices and 5 beta strands, forming an extensive dimer interface. Each dimer has two FMN binding sites, located at the interface, and contributed to by residues from both monomers. One FMN cofactor occupies each active site, binding through non-covalent interactions. The key FMN binding residues (comprising a basic, positively charged residue proximal to the FMN C(2)O, a hydrogen bond donor within 3.5 Å of the FMN N(5) and a basic residue proximal to the FMN phosphate tail) play an important role in modulating the redox potential of FMN [19]. Electron density was found to accommodate one molecule of acetate at the re-face of the FMN in each active site of the crystal structure. Acetate was present in the crystallization buffer and has been reported to act as a competitive inhibitor to nitroreductases [27,28].

### 2.2. Structural Basis for the Improved Activity

To investigate how amino acid substitutions F70A and F108Y improved the nitroreduction of SN33623, metronidazole, and CB1954, we performed a structural comparison with the wild-type enzyme (PDB code 1DS7). Members of the bacterial nitroreductase superfamily to which NfsB belongs [19] are particularly amenable to such comparisons, as their catalysis is primarily driven by their bound FMN cofactors in a manner that does not induce substantial conformation change following substrate binding [21,22,29]. Moreover, the residues that comprise the enzymatic active sites do not partake directly in substrate conversion, but instead play key roles in stabilizing charge on the FMN and positioning the substrates at optimal distances for efficient hydride transfer [24,30]. Thus, a structure derived in the absence of bound substrate can still be considered a suitable model for inferring the roles that key residues play in substrate binding.

The structure of the mutant and the parental enzyme are largely identical (RMSD of Cα atoms between the two structures is 0.32 Å), however some significant conformational changes were noted in the active site (Figure 3). First, the aromatic ring at position F70 is absent in the double mutant. This residue has been proposed to play a role in gating access to the active site [31]. Second, the side chain of N67 adopts an alternate conformation in the double-mutant structure (Figure 4), effectively widening the entrance of the active site’s entrance channel.

Third, in the double-mutant active site, we observe the side chain hydroxyl group of Y108 participates in direct hydrogen bonding to the carboxamide/carboxyl groups of N117 and E102, respectively. Based on the donor–acceptor distances, these hydrogen bonds can be classified as strong and electrostatic (within 2.5–3.2 Å) [32] (Figure 5). This hydrogen bonding may serve to stabilize the active site environment, potentially lowering the free energy of the transition state for nitroreduction of the SN33623, CB1954, and metronidazole intermediates. The Y108 hydroxyl group may also hydrogen bond directly to the fluorine tail of SN33623 to stabilize substrate binding, however in the absence of a substrate-bound structure this is speculative.

Fourth, the aromatic rings at positions F123 and F124 are displaced in the double-mutant structure compared with the wild type and their average B-factors are higher, suggesting greater flexibility. The combined effects of the A70 mutation and N67, F123, and F124 displacements in the mutant structure result in a substantially wider active site channel for substrate entrance, as determined by measuring distances between residues 67 and 123 and residues 70 and 123/124 (Figure 6). Considering the relatively large size of SN33623, it is likely that the widened substrate entrance channel plays a significant role in reducing steric hindrance and thus facilitating SN33623 binding, turnover, and release.

Finally, it should be noted that the F70A and F108Y substitutions may affect the binding and/or orientation of the NAD(P)H cofactor within the active site. Informed by the impacts of similar substitutions in Enterobacter cloacae NfsB [33], Dr Eva Hyde’s team suggested it is likely that F70A and F108Y in *E. coli* NfsB may influence the efficacy of NAD(P)H binding [31]. They then proposed that additional active site substitutions known to enhance CB1954 activity, such as T41L or N71S [24,34], might further destabilize NAD(P)H binding [31]. This provides a plausible explanation as to why we previously observed T41L/F70A/F108Y and T41L/F70A/N71S/F108Y variants of *E. coli* NfsB to be less active with CB1954 than either of the F70A/F108Y or T41L/N71S variants in isolation [17].

We additionally tested whether the AI software AlphaFold2 [35] could predict a structure for *E. coli* NfsB F70A/F108Y that was congruent with our solved crystal structure. We observed that the structure predicted by AlphaFold2 demonstrated a high degree of alignment in the overall backbone architecture with our experimentally determined structure (overall RMSD 0.342 Å on alignment of dimers), however discrepancies were apparent in the exact positioning of residue side chains within the active site (Appendix A). Discrepancies were notably evident for residues A70 and N67, where the AlphaFold2 model more closely resembles the wild type.

Despite multiple crystallization trials employing previously reported protocols [23], attempts to co-crystallize or post-soak the wild-type and the double-mutant proteins with SN33623, CB1954, or metronidazole were unsuccessful and substrate-bound structures were not attained, although an electron density consistent with acetate was sometimes observed in the active site when acetate-containing crystallization buffers were employed. A post hoc analysis of the diffraction data from previously published prodrug-bound *E. coli* NfsB structures (PDB codes 1IDT, 1OON, and 1OO6) similarly revealed insufficient electron density within the active site to justify prodrug placement. In light of these findings, we shifted our focus to computational modeling, using theoretical simulations to explore the impacts of the F70A and F108Y mutations on substrate binding interactions.

### 2.3. Molecular Modeling of Prodrug Binding—E. coli NfsB F70A/F108Y vs. Wild Type

We have previously established a host–guest modeling approach to predict the binding modes of diverse nitroaromatic substrates within the active site centers of wild-type and engineered NfsB enzymes from Vibrio vulnificus [36], as well as NfsA from E. coli [37]. Here, we used this approach to better infer the relative abilities of the wild-type *E. coli* NfsB (henceforth referred to as “wtNfsB”) and mutant *E. coli* NfsB F70A/F108Y (henceforth referred to as “mutNfsB”) enzymes to bind the SN33623 probe, as well as the nitroaromatic prodrugs CB1954 and metronidazole. Our simulations employ a simplified modeling approach that focuses solely on the active center fragment and, as above, assume that the overall enzyme structures remain unchanged upon substrate binding. The residues located beyond 9 Å from N5 of the FMN cofactor (here, F108/Y108) are considered to lie outside the minimal structure and exert no direct influence on the active site centers, although they may play a role in substrate entry. This model allows the replacement of the oxidized FMN cofactor from our solved crystal structure with its reduced and active form (FMNH_2_). It should be noted that the water molecules in the active center are crucial, as they may be displaced by the substrate, and/or mediate hydrogen bonds between key residues and the substrate.

To begin with, electrostatic potential energy maps (EPEMs) were generated for the surfaces of both the active centers and the substrates (see Figure 7). EPEMs offer an insight into the distribution of charge density across molecular surfaces by aiding in understanding how substrates interact within the active center [38]. Figure 7A–C show that the selected substrates possess nitro and imidazole groups with a negative energy charge density. Complementary regions of positive charge density are observed on the surfaces of the active centers of both enzyme forms (Figure 7D,E), particularly on the Lys14 and Lys74 residues due to the protonated amino groups, which facilitate the formation of the complexes. It is noteworthy that the charge densities of the active site centers are similar in both the wtNfsB and the mutant, but the active center of mutNfsB appears to be more spacious due to the substitution of bulky F70 by alanine.

To validate our theoretical simulations, we compared our simulated wtNfsB structure to the crystallographic wtNfsB structure with the reduced FMN cofactor (redFMN) (PDB: 1OO5) obtained by Johansson et al. [23]. Figure 8 depicts that the shape of the reduced FMN cofactor in our simulated structure is slightly bent, consistent with the X-ray structure. Additionally, the position of FMN in the active center is similar. The positions of the Phe70, 124, and Lys14, 74 amino acid residues are slightly altered due to their flexibility in the simulated structure. Overall, though, the presence of similar positions of the cofactor and amino acid residues between our simulations and the X-ray structure PDB: 1OO5 indicates a strong correspondence. Further validation against the prodrug-bound *E. coli* NfsB structures (PDB codes 1IDT, 1OON, and 1OO6) was not pursued owing to the uncertainties with these structures noted above.

We next proceeded to simulate the structures of substrate complexes with the active centers of both wtNfsB and its mutant using our model. These complexes are depicted visually in Figure 9, with the key interactions summarized in Table 2. For SN33623, different orientations of the substrate were observed in the active centers of the wild-type and mutant enzymes (Figure 9A,B). Due to the steric constraints imposed by Phe70, this substrate binds to wtNfsB through H-bonds between Lys14 and the nitro group, and the -NH of Ser39 with N3 of the imidazole moiety. The fluorinated tail of the substrate protrudes out of the active site centers and does not participate in complex formation. These interactions are reversed in mutNfsB, where H-bonds are observed between Lys14 and the imidazole N3, and the Ser39-NH and the nitro groups. This provides a more productive orientation of SN33623 in mutNfsB, because its nitro group is directed toward the N5 of reduced isoalloxazine, with a distance between the N5 atom and the nearest oxygen of -NO_2_ being 3.0 Å (Table 2). In contrast, this distance is 6.6 Å in wtNfsB. These theoretical results are consistent with the previously reported half-maximal inhibitory concentrations (IC_50_) values for *E. coli* expressing either wtNfsB or mutNfsB, which were much lower for cells expressing the mutant than cells expressing the wild-type enzyme (nitroreduced SN33623 being antibacterial at high concentrations) (Table 2).

CB1954 binds in the active center of wtNfsB, forming an H-bond between its amide group and Lys14, whereas its 2-nitro group forms an H-bond with Ser39 (Figure 9C). In contrast, Lys14 in mutNfsB forms H-bonds with both the amide and 2-nitro group of CB1954 (Figure 9D). This provides distances of 4.8 Å and 3.0 Å between the N5 of reduced isoalloxazine and the O atom of the 2-nitro group in wtNfsB and mutNfsB, respectively. *E. coli* NfsB is known to reduce CB1954 at either the 2-NO_2_ or the 4-NO_2_ position (in contrast with *E. coli* NfsA, which reduces CB1954 exclusively at the 2-NO_2_ position) [39]. The orientation of CB1954 in the active site center of wtNfsB in our model ensures the reduction of the nitro group at the 2-NO_2_ position. However, the reduction of the 4-NO_2_ position of CB1954 appears more favorable in the mutNfsB than wtNfsB.

For metronidazole, similar orientations to SN33623 are observed, consistent with the shared 5-nitroimidazole scaffold of these substrates. In wtNfsB, the substrate binds through the interaction of Lys14 with the nitro group and the hydroxy group of the aliphatic sidechain with Lys74 and the 2-carbonyl group of isoalloxazine (Figure 9E). In mutNfsB, the nitro group of metronidazole forms an H-bond with Ser39 and the hydroxyl of the aliphatic chain interacts with Lys14 (Figure 9F). This reduces the distance between the nitro group and N5 of the isoalloxazine from 6.9 Å to 3.0 Å in the mutant enzyme.

Overall, our data indicate that, in each case, the general increase in reactivity observed for mutNfsB relative to the parental enzyme can be rationalized in terms of a decreased distance between the oxygen of the nitro group of the oxidant substrate and N5 of the reduced isoalloxazine of FMN.

## 3. Materials and Methods

### 3.1. Bacterial Strains, Media, and Growth Conditions

*E. coli* BL21 strains were grown in Lysogeny broth (LB) medium at 37 °C with agitation (200 rpm) or on LB agar plates at 37 °C.

### 3.2. Cloning, Expression, and Purification of E. coli NfsB F70A/F108Y

The gene encoding *E. coli* NfsB F70A/F108Y was ordered pre-synthesized from Twist Bioscience (San Francisco, CA, USA). The gene fragment was cloned into pET28a+ (Novagen, Merck, Darmstadt, Germany), which expresses a His_6_ tag at the C-terminus. The expression construct was used to transform *E. coli* BL21 cells and grown in LB medium containing 50 µg mL^−1^ of kanamycin at 37 °C. After induction with 0.5 mM of isopropyl-beta-D-thiogalactoside (IPTG), the culture was incubated for a further 24 h at 18 °C. The culture was harvested by centrifugation at 2600× *g* for 20 min. The nitroreductase was purified using nickel affinity chromatography, where the bound protein was eluted over a two-step wash and elute method, involving an initial wash with 60 mM of imidazole to remove loosely bound non-target proteins and a second wash with 1 M of imidazole to elute mutant *E. coli* NfsB F70A/F108Y (mutNfsB). For further purification, size exclusion chromatography using a Superdex S200 10/300 GL column (GE Healthcare, Uppsala, Sweden) connected to an ÄKTA BasicTM FPLC system was performed using a 40 mM Tris-HCl pH 7.0, 200 mM NaCl buffer. Fractions containing the target protein were assessed for purity by SDS-PAGE analysis and quantified by their 280 nm absorbance trace.

### 3.3. Crystallization

For crystallization, *E. coli* NfsB F70A/F108Y was concentrated to 7 mg/mL in a buffer consisting of 40 mM Tris-HCl pH 7.0 and 200 mM NaCl. Initial protein crystallization trials employed large-scale crystallization screens covering 288 conditions (Hampton Research screening trays Index, PEGRx1 and Crystal Screen) and used the sitting drop vapor diffusion method in 96-well plates at 291.15 K (18 °C). Nitroreductase protein solution (100 nL) was mixed with precipitant solution (100 nL) and equilibrated beside a 100 µL reservoir of precipitant solution. Further crystallization screens employed hanging-drop vapor diffusion in 24-well VDX plates (Hampton Research, Aliso Viejo, CA, USA) at 291.15 K (18 °C). Crystals formed within two weeks by mixing 1 µL of a 7 mg/mL protein solution (40 mM Tris-HCl pH 7.0 and 200 mM NaCl) with 1 µL of precipitant solution (0.2 M sodium chloride, 0.1 M sodium acetate trihydate pH 4.0, and 33% (*w*/*v*) polyethylene glycol 8000) and by equilibrating over a 500 µL reservoir of precipitant solution. Crystals were yellow in color, indicating the presence of the bound FMN cofactor in the oxidized state. Crystals were flash-frozen in liquid nitrogen prior to data collection.

### 3.4. X-ray Data Collection

Due to the high polyethylene glycol concentration in this crystallization condition, no additional cryo-protectant soak was required prior to crystal freezing/testing. For data collection, crystals were mounted in a stream of N_2_ gas at 100 K. Data were collected at the macromolecular crystallography beamline (MX1) at the Australian Synchrotron, Melbourne, Australia, using an ADSC Quantum 210r detector at a wavelength of 0.9537 Å using 0.5° oscillation per image with a crystal-to-detector distance of 250 mm. A data set was collected to 1.87 Å resolution from a single crystal. All data processing was conducted using software found within CCP4 program suite 6.4.0 (Collaborative Computational Project, Number 4. 1994) [40]. The data set was indexed and integrated using iMosflm 7.0.9 [41], which was used for integration and space group determination. The integrated and combined reflections were scaled and merged using SCALA [42]. Data were trimmed to a resolution such that the outer shell R_merge_ was <0.9 and the final structure was solved to 1.98 Å using molecular replacement with the starting model 1DS7. The data collection statistics are summarized in Table 1.

### 3.5. Structure Solution and Refinement

The *E. coli* NfsB F70A/F108Y structure was solved to 1.98 Å by molecular replacement using the structure of the parental enzyme (PDB code 1DS7, 60.4% identity) as a model. Molecular replacement was carried out using MOLREP software (version 11.0/22.07.2010/) [43] in the CCP4 suite (Collaborative Computational Project, Number 4, 1994) [40]. Cycles of manual model building were completed using COOT [44] followed by refinement in Refmac5 [45] (CCP4 suite). Coordinate files and structure factors for the full structure have been deposited in the Protein Data Bank with PDB code 8V5B. The refinement statistics are included in Table 1.

### 3.6. Theoretical Simulations

To predict interactions between substrates and the active centers of *E. coli* wild-type NfsB and the mutant, the crystal structures of the wild-type (PDB ID: 1DS7) [22] and the mutant (PDB ID: 8V5B) enzymes from the Protein Data Bank (PDB) were employed. For simulation purposes, the residues within a 9 Å radius of the N5 atom of the flavin mononucleotide (FMN) cofactor were extracted to prepare input files for the fragment of the enzymes with its active centers. The center structures of these enzymes were generated using UCSF Chimera software (v. 1.16, the University of California, San Francisco, CA, USA). The heterocyclic fragment of FMN was replaced with its reduced form (FMNred). All atoms heavier than hydrogen in the residues, except for side chains of amino acids capable of forming intermolecular bonds with the substrates in the center, were restrained. The amino groups of lysines (Lys14 and Lys74) and the carboxylic group of glutamic acid (Glu165) residues were protonated and deprotonated, respectively, to correspond to the protein states at pH 7. Additionally, water molecules present in the original crystal structure were allowed to move freely within the active center. The significance of water molecules in predicting supramolecular host–guest structures has been demonstrated in our previous works [46,47].

The substrates were positioned to facilitate intermolecular bonding between amino acid residues and the substrate within the active center. The obtained structures were relaxed using the molecular mechanics force field (MMFF) method. Subsequently, the structures and the energies with and without the substrate were optimized and computed using the PM6 (Parameterization Method 6) approach. The semiempirical PM6 method has been shown to reproduce the heats of formation and geometries of peptides comparing with their crystal structures [48]. EPEMs of the active centers and the substrates were generated using this PM6 method. The theoretical approach enabled us to predict the structure of the active center with the substrate and calculate the distances involved in host–guest interactions. The energies of substrate binding (*E_in_*) were calculated using Equation (1):*E_in_* = *E*_(*center*−*S*)_ − *E*_(*center*)_ − *E*_(*S*)_(1)
where *E*_(*center*−*S*)_, *E*_(*center*)_, and *E*_(*S*)_ are the heats of formation of the center substrate, the free center, and free substrate, respectively. The energies were computed using the PM6 method. All computational analyses were performed using Spartan’18 software (Spartan’18 for Windows, v. 1.3.0; 1840 Von Karman Avenue, Suite 370, Irvine, CA, USA).

## 4. Conclusions

In this study, we conducted the structural analysis and comparative modeling of an *E. coli* NfsB double mutant to elucidate the impact of F70A and F108Y substitutions on activity with SN33623, CB1954, and metronidazole. The X-ray crystal structure of the double mutant revealed an expanded substrate access channel compared to the wild type, potentially facilitating the entry of bulky substrates, such as SN33623. Additionally, an introduced hydrogen bond resulting from the F70A substitution may play a role in stabilizing the active site of the double mutant, thereby potentially reducing the free energy of the transition state and enhancing turnover rates for each prodrug. The theoretical modeling data reveal differences in substrate orientations and interactions with amino acid residues in the active center between the wild-type and mutant NfsB enzymes, with the mutant models exhibiting reduced distances between the N5-H of isoalloxazine and the nitro group oxygen of each prodrug. Overall, the simulations provide insights into the structural basis of substrate binding and enzyme activity using a more reliable and accurate semi-empirical PM6 method than the molecular mechanics force field. The experimental and theoretical observations highlighted the importance of specific amino acid residues, such as Lys14, Lys74, Ser39, and Phe70, in the substrate recognition and binding. Collectively, these crystallographic and modeling analyses enable us to infer likely molecular mechanisms underpinning the improved activation of SN33623, CB1954, and metronidazole observed for the double mutant. Such insights may serve to inform future nitroreductase engineering and evolution studies, shaping the future application of nitroreductases in biotechnology.

## Figures and Tables

**Figure 1 ijms-25-06593-f001:**
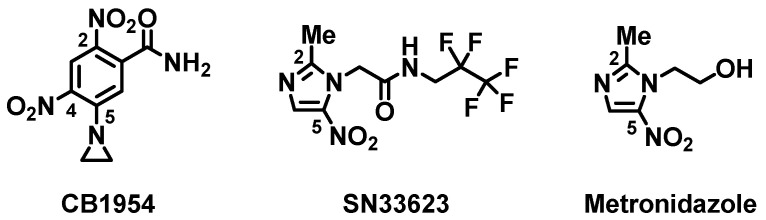
Structures of substrate molecules investigated in this study, with key carbon atoms numbered.

**Figure 2 ijms-25-06593-f002:**
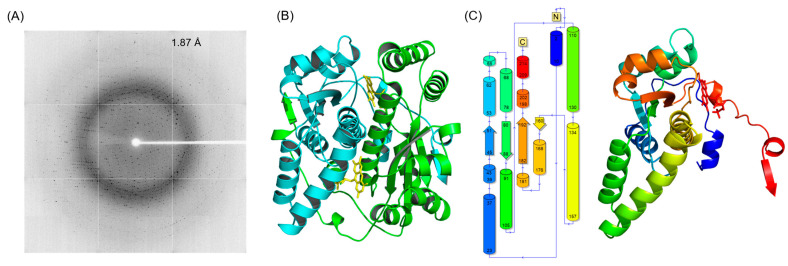
X-ray diffraction and cartoon representations of NfsB F70A/F108Y. (**A**) Representative X-ray diffraction image from a crystal of *E. coli* NfsB F70A/F108Y. The edge of the detector corresponds to a resolution of 1.87 Å. (**B**) Side view of the *E. coli* NfsB F70A/F108Y homodimer. The different monomers are distinguished by teal or green coloring, and FMN molecules (yellow) are illustrated in stick form in each active site. (**C**) Two-dimensional topology structure diagram and three-dimensional structure of the *E. coli* NfsB F70A/F108Y monomer. Numbers represent residue positions. Cylinders represent α-helices, arrows represent β-strands. Spectrum coloring starts at the N terminus of each chain in dark blue, ending at the C terminus of each chain in red. Each dimer consists of two chains, with only a single monomer depicted here.

**Figure 3 ijms-25-06593-f003:**
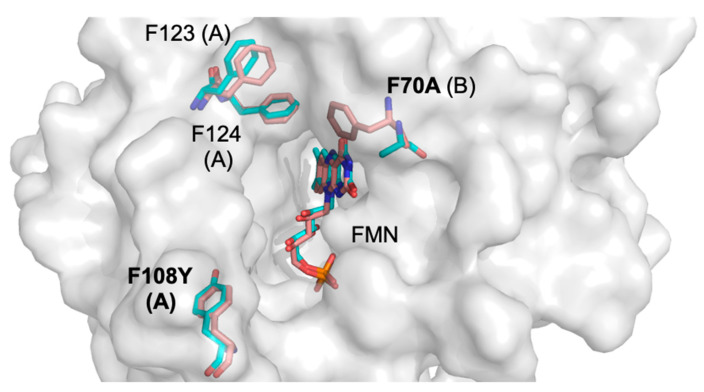
Active site overlay of *E. coli* NfsB F70A/F108Y and the wild-type enzyme. The surface represents the mutant structure. The residues of wild-type (pale red) and mutant (cyan) NfsB exhibiting significant differences and FMN molecules are shown as sticks. Chain identity (A or B) is identified in parentheses after each numbered residue.

**Figure 4 ijms-25-06593-f004:**
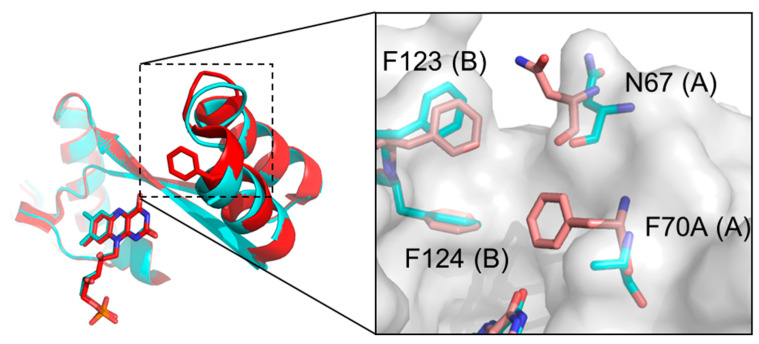
Different N67 conformations adopted in the *E. coli* NfsB wild-type and F70A/F108Y structures. Residues are colored red for the wild type and cyan for the F70A/F108Y mutant. Residues 67, 70, 123, and 124 are shown as sticks. The surface represents the F70A/F108Y structure. Chain identity (A or B) is identified in parentheses after each numbered residue.

**Figure 5 ijms-25-06593-f005:**
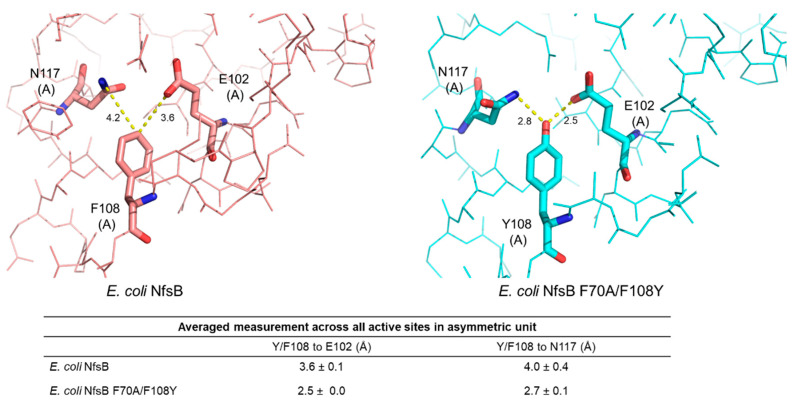
Residue 108 interactions in the active sites of *E. coli* NfsB F70A/F108Y and the parental enzyme. Residues at positions 117, 108, and 102 are shown as sticks and colored red for the parental enzyme or cyan for the mutant. Distance measurements are in Å. The distance between residues 108 and 102 was measured in PyMOL for each active site present in the asymmetric unit of each structure and the average measurement ± SD was calculated. Chain identity (A or B) is identified in parentheses after each numbered residue.

**Figure 6 ijms-25-06593-f006:**
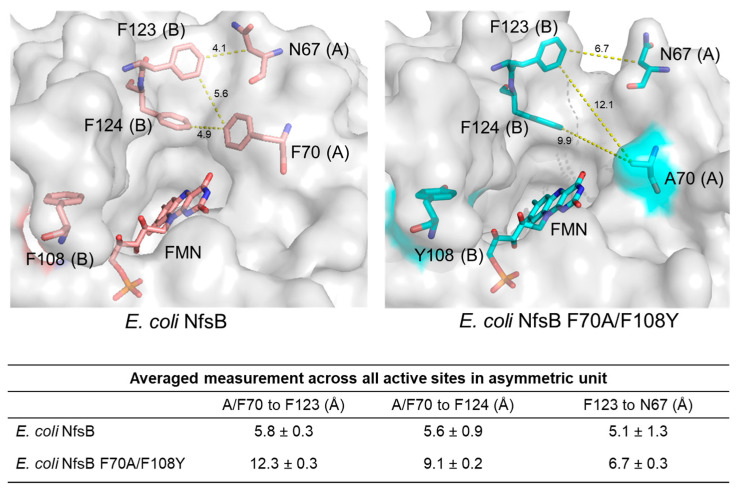
Active site entrance channel width in the *E. coli* NfsB wild-type and F70A/F108Y structures. Residues at positions 67, 70, 108, 123, 124, and FMN are shown as sticks and measurements are in Å. Distances between residues 70 and 123, 70 and 124, and 67 and 123 were measured in PyMOL for each active site in the asymmetric unit of each structure and the average measurement ± SD was calculated. Chain identity (A or B) is identified in parentheses after each numbered residue.

**Figure 7 ijms-25-06593-f007:**
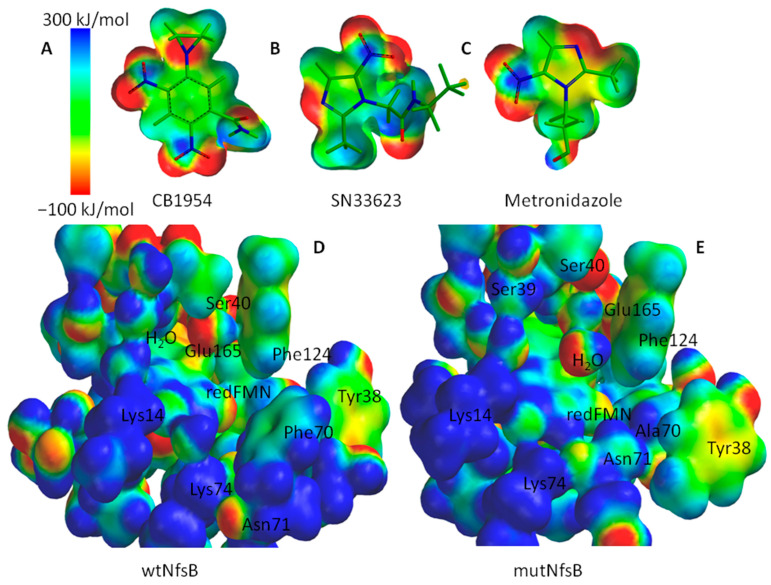
EPEMs of CB1954 (**A**), SN33623 (**B**), metronidazole (**C**), and the active centers of wild-type NfsB (**D**) and NfsB mutant (**E**). Energy range from −100 kJ mol^−1^ to 300 kJ mol^−1^, and an isovalue of 0.01 were used.

**Figure 8 ijms-25-06593-f008:**
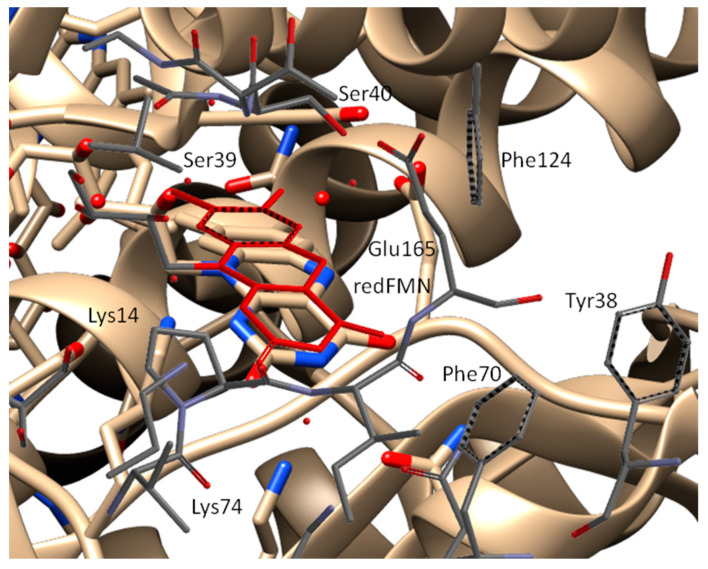
Comparison of the crystallographic and our simulated wild–type NfsB structures. The red moiety is reduced FMN.

**Figure 9 ijms-25-06593-f009:**
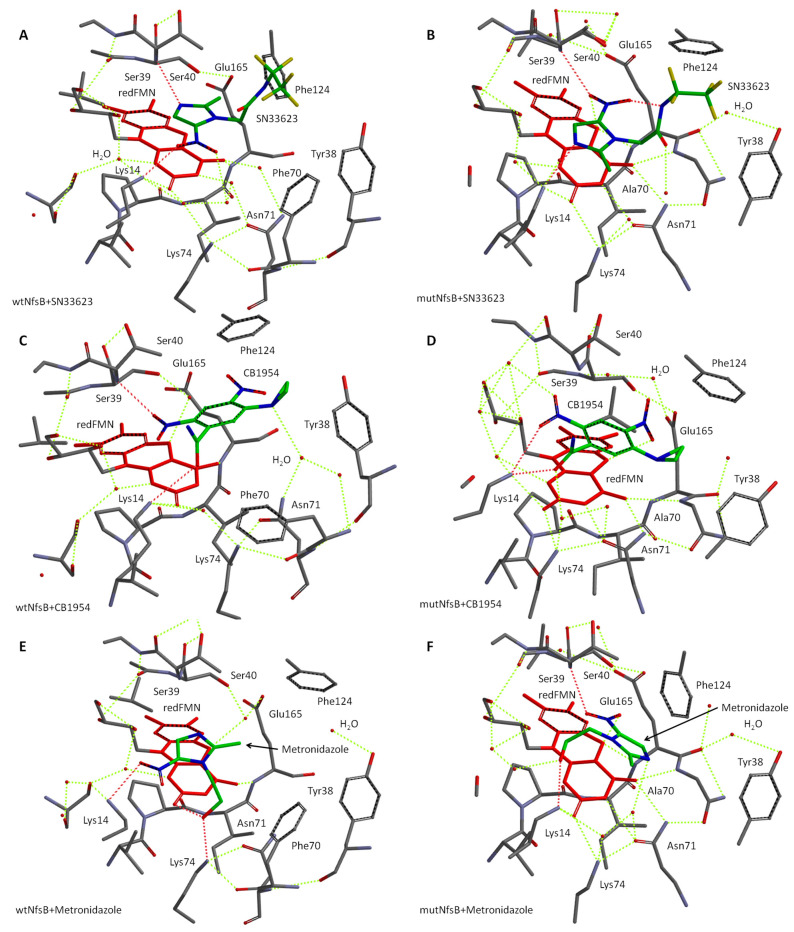
The simulated structures of the active center of wtNfsB in the complex with substrates SN33623 (**A**,**B**), CB1954 (**C**,**D**), and metronidazole (**E**,**F**). The red moiety is reduced FMN; the green molecules are the substrates (red and blue are oxygen and nitrogen atoms, respectively). H-bonds binding the substrate to the active center are shown in red and other H-bonds are shown in yellow dashes. Hydrogen atoms were removed for clarity.

**Table 1 ijms-25-06593-t001:** Data collection and refinement statistics for *E. coli* NfsB F70A/F108Y. Values in parentheses are for the highest-resolution shell.

Wavelength (Å)	0.9537
Space group	P 2_1_ 2_1_ 2_1_
Unit-cell parameters (Å, °)	*a* = 87.22, *b* = 95.59, *c* = 112.31, α/β/γ = 90/90/90
Resolution range (Å)	1.98–87.22 (1.98–2.10)
Measured reflections	469,649 (31,985)
Unique reflections	66,067 (4399)
Multiplicity	7.1
Temperature (K)	100
Matthews coefficient (Å^3^ Da^−1^)	2.45
Solvent content (%)	49.79
No. of molecules in ASU	4
Completeness (%)	100 (100)
Mean *I*/σ(*I*)	9.9 (2.5)
*R*_merge_ ^†^ (%)	15.1 (81.4)
R_work_	0.171
CC(1/2)	0.9615
R_free_	0.219
Protein atoms	6652
Other ions/molecules	14
Number of waters	516
B factors (proteins)	18.164/18.404/16.636/16.618
B factors (waters)	26.403
RMSD	
Bond angles (°)	1.5953
Bond lengths (Å)	0.0103

^†^ *R* _merge_ = $\frac{\sum_{hkl}{\;\sum }_{i}\left|I_{i}\left(hkl\right)-\bar{I}(hkl)\right|}{\sum_{hkl}\sum_{i}I_{i}(hkl)}$, where $I_{i}(hkl)$ is the observed intensity of an individual reflection and $\bar{I}(hkl)$ is the mean intensity of that reflection.

**Table 2 ijms-25-06593-t002:** Key active site interactions from simulations and the previously measured experimental IC_50_ data from [17] for the wild-type and mutant NfsB enzymes with corresponding substrates.

Entry	Enzyme+Substrate	Intermolecular Interactions	d(N5···O) ^a^, Å	−E_in_, kJ mol^−1^	IC_50_, μM
1	wtNfsB+SN33623	Lys14···NO_2_, Ser39-NH···N of imidazole	6.6	63.3	>400
2	mutNfsB+SN33623	Lys14··· N of imidazole, Ser39-NH··· NO_2_	3.0	50.6	41 ± 1
3	wtNfsB+CB1954	Lys14···HNH-CO, Ser39-NH···2-NO_2_	4.8	59.5	380 ± 6
4	mutNfsB+CB1954	Lys14···HNH-CO, Lys14···2-NO_2_	3.0	43.5	110 ± 13
5	wtNfsB+metronidazole	Lys14···NO_2_, Lys74···HO-, FMN C=O···HO-	6.9	85.7	230 ± 39
6	mutNfsB+metronidazole	Ser39-NH··· NO_2_, Lys14···HO-	3.0	20.4	8 ± 3

^a^ Distances were measured between N5 of reduced FMN and the nearest oxygen of NO_2_.

## Data Availability

Structural data associated with this study are available from the Protein Data Bank (PDB code 8V5B). All other data are available upon request.

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
