# Peer review of "Structural Evaluation of a Nitroreductase Engineered for Improved Activation of the 5-Nitroimidazole PET Probe SN33623"

_ijms, 2024, doi:10.3390/ijms25126593_

Round 1
Reviewer 1 Report
Comments and Suggestions for Authors
The manuscript “Structural Evaluation of a Nitroreductase Engineered for Improved Activation of the 5-Nitroimidazole PET Probe SN33623” by Sharrock et al. characterizes the structure of the nitroreductase NfsB mutant F70A/F108Y that enhanced reduction of the theranostic molecule SN33623. The authors describe the 1,98 A resolution X-ray crystallography structure of NfsB F70A/F108Y. This structure was compared with the structure of the WT NfsB (pdb 1DS7) to find that the active site is wider in the structure of the mutant NfsB, facilitating the binding and reduction of SN33623 and other molecules. Although they were not able to solve the structure of NfsB F70A/F108Y bound to any of its substrates, molecular modeling studies showed differences in substrates binding and interactions, explaining the improvements in activity. Thus, this is a simple, well-written article with clear findings that could be worth for publishing in International Journal of Molecular Sicences.
As a minor comment it could be good to discuss if the X-ray crystal structure of the mutant was absolutely required or the new advances in protein structure prediction could have predict exactly the same observed changes.
Author Response
"As a minor comment it could be good to discuss if the X-ray crystal structure of the mutant was absolutely required or the new advances in protein structure prediction could have predict exactly the same observed changes."
We thank the reviewer for this suggestion. This has now been done, and we have added a new supplementary figure (Figure S2) that illustrates the comparison, as well as commenting on this point in the main text.
Reviewer 2 Report
Comments and Suggestions for Authors
Comments:
The article by Abigail et al; reports the evaluation of an engineered nitroreductase enzyme. The article is well written, and the structure of the engineered enzyme is resolved at high resolution to support the findings reported in this article.
The following major comments need to be addressed before considering the manuscript for publication.
1. Page 3, line 91-99, This catalytic mechanism involves the transfer of electrons from a reduced cofactor, -------
Please provide a graphical representation of this mechanism, so that it will be easy for the readers to follow.
2. Figure 2B; Please colour the monomer in different colours, so that it is easy to follow. For eg: monomer A in Cyan and B in light green and show the FMN in yellow.
3. Also please provide a figure for single monomer and combine it with Figure 2C.
4. Figure 8 looks very crowded, it will be great to show the backbone in ribbon representation instead of cartoon.
5. Please check the bond length outliers for the mutated residue A70 in chain A.
Author Response
- Page 3, line 91-99, This catalytic mechanism involves the transfer of electrons from a reduced cofactor, -------
Please provide a graphical representation of this mechanism, so that it will be easy for the readers to follow.
This suggestion has been adopted, and we have provided the requested illustration as a new supplementary figure (Figure S1) that is referred to in the main text.
- Figure 2B; Please colour the monomer in different colours, so that it is easy to follow. For eg: monomer A in Cyan and B in light green and show the FMN in yellow.
- Also please provide a figure for single monomer and combine it with Figure 2C.
These suggestions have also been adopted, and we have provided a new Figure 2 that includes the recommended updates to panels 2B and 2C.
- Figure 8 looks very crowded, it will be great to show the backbone in ribbon representation instead of cartoon.
We tried a range of alternative styles, without generating an improved figure. We therefore retained the original figure style, but did our best to ‘declutter’ it – in particular, moving the residue labels and other text to ensure black-on-white contrast wherever possible. We believe that this has addressed the reviewer’s core concern.
- Please check the bond length outliers for the mutated residue A70 in chain A.
The bond lengths are correct as reported. Whilst the observed C-O bond in Chain A residue 70 is slightly longer than the ideal bond length of 1.23 Å, it is supported by the density and is similar in length to the C-O bonds of residue 70 across the other chains of the asymmetric unit (measuring 1.31 Å, 1.26 Å and 1.30 Å, respectively). Distortion in the C-O bond length of residue 70 could result from crystal packing effects, or these bond lengths may be influenced by their chemical environment, such as the presence of the adjacent FMN cofactor.